# Understanding knowledge, attitude and perception of Rift Valley fever in Baringo South, Kenya: A cross-sectional study

**Tatenda Chiuya**[1]*, **Eric M. Fevre**[2,3], **Sandra Junglen**[4], **Christian Borgemeister**[1]

**1** Centre for Development Research (ZEF), University of Bonn, Bonn, Germany, **2** International Livestock Research Institute (ILRI), Nairobi, Kenya, **3** Institute of Virology, Charité - Universitätsmedizin Berlin, Corporate Member of Free University Berlin, Humboldt-University Berlin and Berlin Institute of Health, Berlin, Germany, **4** Institute of Infection, Veterinary and Ecological Sciences, University of Liverpool, Leahurst Campus, Neston, United Kingdom

\* tchiuya@uni-bonn.de, tatendachiuya@gmail.com

**Data Availability Statement:** The dataset generated and analysed in this study is available on the University of Liverpool Research Data

## Abstract

Rift Valley fever (RVF) is a mosquito-borne viral hemorrhagic disease that affects humans and livestock. In Kenya, the disease has spread to new areas like Baringo County, with a growing realization that the epidemiology of the virus may also include endemic transmission. Local knowledge of a disease in susceptible communities is a major driver of prevention and control efforts. A cross-sectional survey using a semi-structured questionnaire was conducted in five locations of Baringo South that had reported RVF cases during the last outbreak, to determine the knowledge, attitude and perception of the predominantly agro-pastoralist community to RVF. Knowledge of RVF clinical signs, transmission, risk factors and prevention all contributed to the total knowledge score. Additionally, the respondents' attitude was based on their awareness of the threat posed by RVF and preparedness to take appropriate measures in case of suspected infection. Out of the 300 respondents, 80% had heard about the disease, however, only 9.6% attained at least half of the total knowledge score on RVF. Nevertheless, 86% recognized the threat it posed and knew the appropriate action to take in suspected human and livestock cases (positive attitude). Factors significantly associated with a better knowledge of RVF included higher education level, being Maasai, higher socio-economic index, old age and history of RVF in household members and livestock. Being Maasai and a higher socio-economic index were significantly associated with a positive attitude. The low level of knowledge exhibited by the respondents could be due to progressive loss of interest and information associated with a prolonged inter-outbreak period. This calls for regular awareness campaigns. More emphasis should also be put on educating communities on the role played by the mosquito vector in the epidemiology of RVF. The most promising routes of disseminating this information are radio and community gatherings.

Catalogue with the following DOI: https://doi.org/10.17638/datacat.liverpool.ac.uk/2387.

**Funding:** This study was supported by the German Research Foundation (DFG), through funding for the project "Future Infections" as part of the Collaborative Research Center "Future Rural Africa" (TRR 228/1 to CB and SJ). This study also received partial support from the CGIAR One Health initiative "Protecting Human Health Through a One Health Approach", which was supported by contributors to the CGIAR Trust Fund (https://www.cgiar.org/funders/) (to EMF). The funders had no role in study design, data collection and analysis, decision to publish, or preparation of the manuscript.

**Competing interests:** The authors have declared that no competing interests exist.

**Abbreviations:** RVF, Rift Valley fever; RVFV, Rift Valley fever virus; CI, confidence interval; WHO, World Health Organisation; KNBS, Kenya National Bureau of Statistics; NACOSTI, National Commission of Science, Technology and Innovation; ILRI, : International Livestock Research Institute; IREC, Institutional Research Ethics Committee; MCA, multiple correspondence analysis; PCA, principal component analysis; OR, odd ratio; AUC, area under curve; ROC, receiver operating curve; ZEF, Center for Development Research.

## Introduction

Rift Valley fever (RVF) is a zoonotic disease caused by the Rift Valley fever virus (RVFV) (genus: *Phlebovirus*, family: *Phenuiviridae*) [1]. The disease, which was first reported in the Rift Valley region of Kenya [2] is endemic in sub-Saharan Africa but outbreaks have occurred on the Indian Ocean island of Mayotte and the Arabian Peninsula as well [3–5]. It is characterized by high mortality in young livestock and '*abortion storms*' in pregnant animals (Pepin et al., 2010 [6]). In humans, the disease usually manifests as a self-limiting acute febrile illness, but may progress to hepatitis, encephalitis and/or hemorrhagic fever in rare cases [7]. The occurrence of RVF outbreaks is associated with persistent rainfall and flooding in dry areas, increase in abundance of *Aedes* spp. (Diptera: Culicidae), presence of susceptible livestock and the virus itself [8]. However, in regions with abundant rainfall, transmission of the virus can be endemic [9]. Humans are infected through contact with infected animals, their products and secretions and also through mosquito bites [9, 10].

Ever since the first reported outbreak in 1931, eleven national RVF outbreaks have occurred in Kenya with the most widespread being in 2007/2008 [11]. The number of counties affected have drastically increased, signifying increased potential for transmission in most parts of the country [12]. Climate change, ubiquitous presence of competent vectors and susceptible ruminants and movement of viraemic animals are the major drivers of RVF introduction into new areas [13]. Recently, smaller, spatially limited outbreaks were reported in 2014, 2018 and 2021 [14]. Baringo County is one of the recently reported hotspots for RVF, with several locations and villages being affected in the last major outbreak of 2007/2008 during which it reported the highest livestock and human cases (85 human cases and 5 deaths) [15]. In Baringo, there are high cattle, sheep and goat populations, and regular flooding events associated with Lakes Baringo and Bogoria. There are also clay-rich, impermeable soils with high surface water retention properties (solanchak soils) [16]. Considering the presence of these factors, the virus is likely to pose a constant threat in the County.

Livestock vaccination remains one of the most important methods of preventing outbreaks in Kenya. However, it has to be done consistently and before the outbreak to be effective. In addition to the medical/veterinary approach, an important but often understated mitigation tool before and during outbreaks is education campaigns and establishment of effective communication chains. Awareness has the ability to lower the number of infected individuals and the peak of an epidemic [17, 18]. An inequitable flow of information was one of the factors attributed for a poor response in Kenya during the 2007/2008 RVF outbreak [19]. Despite the devastating effects of RVF outbreaks in susceptible regions, studies show that the knowledge on the disease has only marginally improved among affected communities [15, 20–22].

Long inter epidemic periods may lead to people losing interest and the acquired knowledge on RVF. Additionally, previous surveys in Baringo County have not thoroughly investigated community knowledge on RVF vectors, their behavior and ecology in light of climate change and the massive presence of invasive plants such as *Prosopis* spp. (Fabaceae), *Parthenium* spp. (Asteraceae), *Opuntia* spp. (Cactaceae) and *Lantana camara* L. (Verbenaceae) in the area. These plants provide suitable micro-habitats and oviposition sites for mosquitoes, are a source of sugar, and are thought to influence vectoral capacity of mosquitoes for arboviruses [23]. It is possible that inadequate information is being collected by surveys to assist in designing effective awareness campaigns. This study was therefore carried out to determine the knowledge of the agro-pastoralist communities in Baringo County on RVF, their attitudes, perception and coping strategies. Socio-economic and demographic data were collected to assess their influence on community knowledge of RVF.

## Methods and materials

### Ethics statement

The questionnaire interviews were carried out after approval by the International Livestock Research Institute (ILRI) Institutional Research Ethics Committee (ref ILRI-IREC2022-25) licensed by the National Commission for Science, Technology and Innovation (NACOSTI: License No: NACOSTI/P/22/19512) in Kenya. Permission to carry out sampling was also obtained from the Deputy County Commissioner's office, area chiefs and village elders. Oral and informed consent was sought from the respondents at households before they were enrolled into the study.

### Study site

The study was carried out in Baringo County which is found in the Kenyan Rift valley and consists of both highland and lowland areas (Fig 1). Annual precipitation in the highlands has

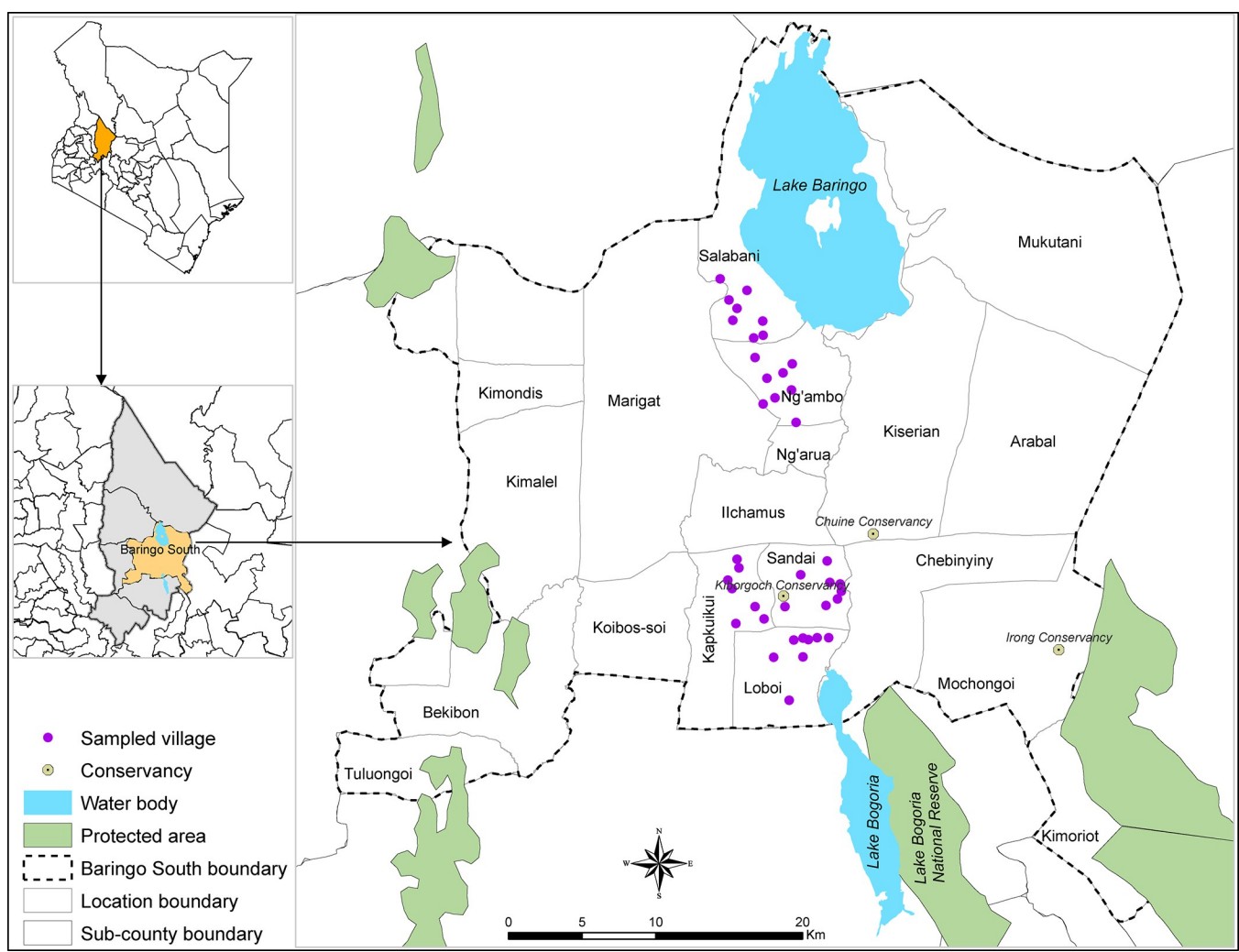

**Fig 1. A map of Baringo County, Baringo South sub-County and the locations (villages) selected for the questionnaire survey.** Water bodies, country and county boundary data was downloaded from the World Resources Institute (https://www.wri.org/resources/data_sets) [25]. The map was developed using ArcGIS Software Version 10.2 (http://desktop.arcgis.com/en/arcmap) [26].

been recorded between 1,000–1,500 mm and 300–700 mm in the low-lying areas, with peak precipitation experienced in April and November [24]. The climate is hot and arid with temperatures ranging from 10˚C in the highlands to 35˚C in the lowlands. Most of the vegetation is represented by tree species such as *Prosopis juliflora* (Sw.) DC. (Fabaceae), *Opuntia* spp., and *Acacia* spp. (Fabaceae) while other species such as *Parthenium* spp. are mostly found in agricultural plots after harvest and along irrigation channels. The County is inhabited by Tugens who are a Kalenjin sub-group and the Ilchamus who are a Maasai sub-group. The major economic activity is agro-pastoralism with crops dependent on rainfall or grown close to the shores of Lake Bogoria or Lake Baringo to facilitate flood irrigation. Other income generating activities include charcoal burning, conservation, tourism and apiculture. The Ilchamus are mostly livestock keepers and predominantly inhabit the lowlands while the Tugens tend to be crop farmers and inhabit the highlands [15].

### Study design, sampling and data collection

This was a cross-sectional survey using a questionnaire (S1 Questionnaire) with both closed and open-ended questions administered to agro-pastoralists in five locations of Baringo South sub-County, namely Sandai, Loboi, Kapkuikui, Salabani and Ng'ambo. These were purposively selected because (i) most of the RVF cases reported in Baringo County during the 2007/2008 outbreak were reported in these locations, (ii) they are associated with three community conservancies and one national park, (iii) two lakes (Lakes Baringo and Bogoria) and (iv) the area is infested by a variety of invasive alien plants. The required total number of respondents was calculated using the formula for sample size determination [27]. Considering previous knowledge level of RVF to be 20% in Baringo County [20] and a non-response rate of 10%, a total of 271 respondents were required, with eventually 300 respondents interviewed. In each of the locations, eight villages were randomly selected. Each location is made up of 2 sub-locations, therefore each sub-location contributed 4 villages.

Complete households' lists were obtained from village heads through the location chiefs and used to select respondents using stratified random sampling. The total number of respondents was proportionally constituted based on the number of households in each location. According to available recent census data, there are 296 households in Kapkuikui, 672 in Loboi, 954 in Ng'ambo, 967 in Salabani and 707 in Sandai giving a total of 3596 households in the five locations [28]. Therefore, respondents from Kapkuikui constituted 8.2% of the study population while those from Loboi, Ng'ambo, Salabani and Sandai made up 18.7%, 26.5%, 26.9%, and 19.7% of the study population respectively. Trained Tugen and Ilchamus speaking enumerators carried out face to face interviews in the selected households in October of 2022. At each household the head or another person from that household above the age of 18 were interviewed and data were recorded using ODK collect v2022.4.0 software which enabled us to monitor the submission of the enumerators daily for quality control.

### Inclusivity in global research

Additional information regarding the ethical, cultural, and scientific considerations specific to inclusivity in global research is included in the supporting information (S1 Text).

### Data management

All data was downloaded from the ODK server into MS Excel files. The data were cleaned to remove redundant information before analysis.

## Independent variables

The independent variables in this study included age of the household head, gender, ethnicity, education level and history of RVF in the household or livestock. Livestock ownership, was converted into tropical livestock units [29]. The mean total tropical livestock units were calculated. A socio-economic index was calculated using scores generated from variables such as house roofing type, floor type, wall material, ownership of electronics, energy used by the household for cooking and lighting, source of drinking water in the household and type of toilet used by the household. Multiple correspondence analysis (MCA) was used to derive the socio-economic index in R (version 4.2.2) using the *FactoMineR* and *factoextra* functions for visualization [30, 31]. Multiple correspondence analysis is a reduction method that is an extension of Principal component (PCA) analysis (for continuous data). It is used to analyze relationships in multi-dimensional categorical population data by converting it into fewer dimensions that explain much of the variation within a study population. It reveals the quantity and quality of the contribution of each variable to these dimensions on a numerical scale. The weighted contribution is then used to generate a numerical variable for each individual [30, 31]. In this study the first dimension (factor) derived from the MCA captured most of the variation in the component variables and was used to generate the underlying household socio-economic variable (S2 Text).

## Dependent variables

The overall knowledge on RVF was sub-scaled into knowledge of transmission, clinical signs, risk factors and preventative practices. A knowledge score was generated for each of these subscales where one point was given for a correct answer while a zero was awarded for a wrong response/no response/ "*I don't know*". Where a Likert scale was used one point was given when agree/strongly agree and disagree/strongly disagree were used appropriately and zero when not. Knowledge of transmission generated a total of 9 points, knowledge of clinical signs had 15 points, knowledge of risk factors a total of 9 points and knowledge of preventative measures a total of 10. These scores were generated by considering the most important routes of transmission, clinical signs, risk factors and preventative methods of RVF. The same scoring method was utilized for generating a score for attitude (4 points). Attitude was determined from how the respondents perceive the threat (serious, mild, not important etc.) posed by RVF and the action they would take in suspected human and livestock cases (seek professional medical/veterinary advice, use traditional herbs, do nothing etc.). Attitude is an important factor in the success of community-based awareness programmes and decision-making. Communities with a positive attitude are more likely to follow and institute prevention measures during RVF outbreaks compared to those with a negative attitude. A matrix of correct answers (*successes*) and wrong answers (*failures*) was established for each sub-scale variable and also the overall knowledge [21].

## Data analysis

The matrix of *successes* and *failures* was used as the dependent variable initially for univariate analysis and then multivariate analysis for the total knowledge of RVF and attitude scores. Participants who scored at least 50% of the total scores were deemed to have better knowledge and attitude to RVF respectively. A logistic regression model was fitted to the data to determine the association between the independent and dependent variables. All the factors with *p*-value less than 0.2 in the univariate analysis were included in the multivariate model for both overall knowledge and attitude. The analysis was carried out in R (version 4.2.2) using the *stepAIC* function (*reverse*) to derive the final knowledge and attitude models

considering potential confounders and interactions. Multicollinearity among all the variables was checked using the generalised variance inflation factors (GVIFs) for both continuous and categorical variables where a value below 10 is considered acceptable [32]. The Hosmer-Lemeshow goodness of fit test was carried out on the final models to assess if they fitted the observed data adequately. In this test the null hypothesis is that there is no difference between the observed data and that predicted by the model. Receiver operating curves (ROC) with area under the curve (AUC) values were used to show the models' predictive ability. Area under the curve values between 0.7 and 0.8 are considered acceptable while 0.8 to 0.9 is excellent, and more than 0.9 outstanding [33]. A *p*-value of less than 0.05 was considered significant in the final models.

## Results

### Socio-demographic characteristics of households' heads

Respondents at 300 households from five locations in Baringo South sub-County were interviewed during the survey. Among the respondents 72.7% (218/300) were household heads while the remaining 27.3% (82/300) were either a wife (73), son (4), sister (1), grandchild (1), daughter in-law (1), daughter (1) or brother (1) to the household head. Because of proportional sampling 26.7% (80/300) of the respondents resided in Salabani and Ng'ambo locations each, while 19.7% (59/300) were from Sandai, 19% (57/300) from Loboi and 8% (24/300) from Kapkuikui (Table 1). Across all five locations the majority of the household heads were male 69% (207/300). Most (23.7%; 71/300) of the household heads were aged between 30–39 years while 49.7% (149/300) had an education up to primary level. The majority (77%; 231/300) were married and 60.3% (181/300) belonged to a protestant religion. Only 35% (104/300) of the households had a ranking ≥ 4 tropical livestock units (the mean tropical livestock units of the selected households were 3.6) (Table 1). The major economic activity was farming as practiced by 83% (250/300) of the interviewed households. The demographics also show that only two major ethnic groups (Maasai and Kalenjin) reside in these locations with minimal intermixing. Households were furthest from social amenities in Salabani compared to the other studied four locations according to the mean distances to a school/health center. Households in Sandai, Loboi and Kapkuikui were closer to a conservation area compared to those in Salabani and Ng'ambo (Table 1). Overall, 54.7% (164/300) of the households interviewed were classified below the mean socio-economic index. In the five locations, 12% (36/300) reported having a member of the household that had suffered from RVF previously, while 23% (69/300) reported having livestock that had suffered from RVF before. These reported incidences of RVF were highest in Ng'ambo followed by Salabani, Kapkuikui, Loboi and Sandai in decreasing order for both human and livestock cases.

### Knowledge of Rift Valley fever

The most popular methods of disseminating information about RVF were radio and television, followed by community gatherings and veterinary/health professionals (Fig 2). More respondents had heard about RVF affecting animals 264/300 (88%) than humans 252/300 (84%) while 80% (240/300) had heard about the disease affecting both humans and animals. The majority of the respondents 223/300 (74.3%) were able to mention that the disease is zoonotic. Most of the respondents had heard about RVF in animals from veterinary professionals while knowledge on the disease in humans originated from health professionals. The '*Other*' category included respondents who had heard about RVF from government officials, elders and at school (Fig 2). Overall, only 9.6% of the interviewed households managed to score at least half of the total scores which shows low knowledge on RVF. While knowledge on the risk factors of

**Table 1. Socio-economic and demographic characteristics of household heads.**

| Variable | Sandai | Loboi | Kapkuikui | Salabani | Ng'ambo | Total |
|---|---|---|---|---|---|---|
| | n = 59 | n = 57 | n = 24 | n = 80 | n = 80 | N = 300 |
| | n (%) | n (%) | n (%) | n (%) | n (%) | n (%) |
| *Gender* | | | | | | |
| Male | 45 (76.3) | 40 (70.2) | 19 (79.2) | 59 (73.7) | 44 (55) | 207 (69) |
| Female | 14 (23.7) | 17 (29.8) | 5 (20.8) | 21 (26.3) | 36 (45) | 93 (31) |
| *Age* | | | | | | |
| 19–29 | 6 (10.2) | 5 (8.8) | 5 (20.8) | 17 (21.3) | 18 (22.5) | 51 (17) |
| 30–39 | 20 (33.9) | 16 (28.1) | 5 (20.8) | 11 (13.8) | 19 (23.8) | 71 (23.7) |
| 40–49 | 10 (16.9) | 14 (24.6) | 4 (16.7) | 24 (30) | 10 (12.5) | 62 (20.7) |
| 50–59 | 9 (15.3) | 8 (14) | 6 (25) | 18 (22.5) | 10 (12.5) | 51 (17) |
| 60 + | 14 (23.7) | 14 (24.6) | 4 (16.7) | 10 (12.5) | 23 (28.8) | 65 (21.7) |
| *Education level* | | | | | | |
| None | 10 (16.9) | 12 (21.1) | 4 (16.7) | 12 (15) | 30 (37.5) | 68 (22.7) |
| Primary | 25 (42.4) | 32 (56.1) | 8 (33.3) | 46 (57.5) | 38 (47.5) | 149 (49.7) |
| Secondary | 19 (32.2) | 11 (19.3) | 6 (25) | 15 (18.8) | 6 (7.5) | 57 (19) |
| Tertiary | 4 (6.8) | 2 (3.5) | 4 (16.7) | 6 (7.5) | 6 (7.5) | 22 (7.3) |
| University | 1 (1.7) | 0 (0) | 2 (8.3) | 1 (1.3) | 0 (0) | 4 (1.3) |
| *Marital status* | | | | | | |
| Married | 42 (71.2) | 43 (75.4) | 18 (75) | 69 (86.3) | 59 (73.8) | 231 (77) |
| Divorced | 2 (3.4) | 2 (3.5) | 0 (0) | 1 (1.3) | 6 (7.5) | 11 (3.7) |
| Widowed | 8 (13.6) | 8 (14) | 4 (16.7) | 7 (8.8) | 11 (13.8) | 38 (12.7) |
| Single | 7 (11.9) | 4 (7) | 2 (8.3) | 3 (3.8) | 4 (5) | 20 (6.7) |
| *Religion* | | | | | | |
| Non-religious | 1 (1.7) | 3 (5.3) | 0 (0) | 3 (3.8) | 4 (5) | 11 (3.7) |
| African traditional religion | 2 (3.4) | 0 (0) | 1 (4.2) | 1 (1.3) | 5 (6.3) | 9 (3) |
| Catholic | 31 (52.6) | 33 (57.9) | 11 (45.8) | 15 (18.8) | 8 (10) | 98 (32.7) |
| Muslim | 0 (0) | 0 (0) | 0 (0) | 0 (0) | 1 (1.3) | 1 (0.3) |
| Protestant | 25 (42.4) | 21 (36.8) | 12 (50) | 61 (76.3) | 62 (77.5) | 181 (60.3) |
| *Tropical livestock units* | | | | | | |
| < 4 | 37 (62.7) | 33 (57.9) | 11 (45.8) | 51 (63.8) | 64 (80) | 196 (65.3) |
| ≥ 4 | 22 (37.3) | 24 (42.1) | 13 (54.2) | 29 (36.3) | 16 (20) | 104 (34.7) |
| *Socio-economic index* | | | | | | |
| High | 38 (64.4) | 28 (49.1) | 14 (58.3) | 32 (40) | 24 (30) | 136 (45.3) |
| Low | 21 (35.6) | 29 (50.9) | 10 (41.7) | 48 (60) | 56 (70) | 164 (54.7) |
| *Economic activity* | | | | | | |
| Salaried employment | 3 (5.1) | 1 (1.8) | 2 (8.3) | 5 (6.3) | 2 (2.5) | 13 (4.3) |
| Farming | 53 (89.8) | 50 (87.7) | 18 (75) | 65 (81.3) | 64 (80) | 250 (83.3) |
| Other | 3 (5.1) | 6 (10.5) | 4 (16.7) | 10 (12.5) | 14 (17.5) | 37 (12.3) |
| *Native* | | | | | | |
| NO | 8 (13.6) | 18 (31.6) | 7 (29.2) | 5 (6.3) | 4 (5) | 42 (14) |
| YES | 51 (86.4) | 39 (68.4) | 17 (70.8) | 75 (93.8) | 76 (95) | 258 (86) |
| *Ethnicity* | | | | | | |
| Kalenjin | 59 | 56 | 24 | 4 | 0 | 143 |
| Maasai | 0 | 0 | 0 | 76 | 79 | 155 |
| Other | 0 | 1 | 0 | 0 | 1 | 2 |
| *Previous experience with RVF* | | | | | | |
| Household member | 3 (5.1) | 3 (5.3) | 2 (8.3) | 8 (10) | 20 (25) | 36 (12) |

*(Continued)*

**Table 1.** (Continued)

| Variable | Sandai | Loboi | Kapkuikui | Salabani | Ng'ambo | Total |
|---|---|---|---|---|---|---|
| | **n = 59** | **n = 57** | **n = 24** | **n = 80** | **n = 80** | **N = 300** |
| | **n (%)** | **n (%)** | **n (%)** | **n (%)** | **n (%)** | **n (%)** |
| Livestock | 5 (8.5) | 8 (14) | 4 (16.7) | 24 (30) | 28 (35) | 69 (23) |
| *Proximity (mean distance)* | | | | | | |
| Health center | 2.4 km | 3 km | 3.9 km | 9.1 km | 2.2 km | - |
| Primary school | 1.5 km | 1.4 km | 1.6 km | 2.2 km | 1.2 km | - |
| Secondary school | 2.7 km | 2.8 km | 3.8 km | 4.2 km | 5.5 km | - |
| Conservation area | 4.3 km | 2.6 km | 2.9 km | 33 km | 43.8 km | - |

RVF was high (64.3%), it was generally low for transmission (15.3%), prevention (10%) and clinical signs (9%). Most of the respondents 204/300 (68%) implicated grazing spaces as the major source of the RVFV followed in decreasing order by watering places (lakes) 79/300 (26.3%) and bushy areas 67/300 (22.3%).

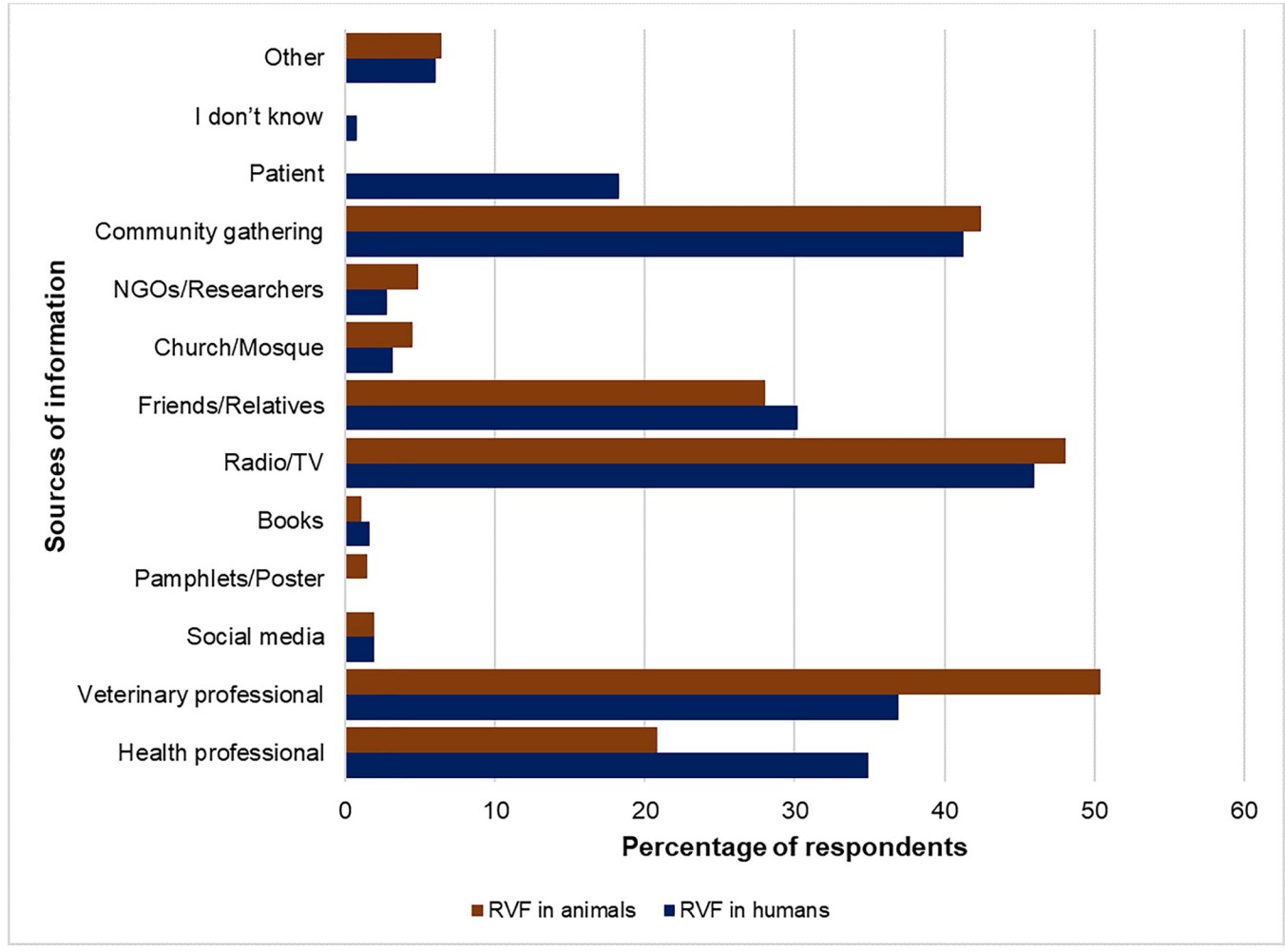

**Fig 2. Sources of knowledge on Rift Valley fever.**

## Knowledge of Rift valley fever transmission routes

RVF transmission in animals was mostly attributed to contact with infected animals 108/300 (36%) at pasture, mosquito bites 87/300 (29%) and *via* water contaminated by animals shedding the virus at watering points 73/300 (24.3%). Transmission to humans was thought to mostly occur through a combination of eating meat and drinking milk from infected animals 99/300 (33%) while eating meat from infected animals alone was mentioned by 48/300 (16%) of the respondents. Despite the respondents mentioning a variety of combinations as shown in Table 2 its noteworthy that consumption of undercooked animal products featured prominently. Other transmission routes of RVF in animals falling out of the classes indicated in the table included; tick and tsetse fly bites, airborne and through "*eating a specific grass*".

None of the respondents were able to name the mosquito species responsible for transmitting RVF as belonging to the *Aedes* spp. However, 15% (45/300) of the respondents were able to correctly describe the appearance of the mosquito as black and white dotted or '*zebra like*' which fits the description of *Aedes* spp. mosquitoes. Most of the respondents reported being bitten by mosquitoes more in the rainy season 276/300 (92%), during the evening 191/300 (63.7%) and indoors 232/300 (77.3%). Biting places that were mentioned included bushes, irrigation plots, conservancy areas and grazing lands. With our interest in the role of invasive plants on mosquito bionomics, the respondents implicated *Prosopis* sp. (47%; 141/300) and *Opuntia* sp. (4.7%; 14/300) as the vegetation promoting mosquito proliferation in the area.

**Table 2. Transmission routes of Rift Valley fever as reported by respondents by location.**

| RVF transmission route | Frequency (Proportion %) | | | | | |
|---|---|---|---|---|---|---|
| **Animals** | Kapkuikui (n = 24) | Loboi (n = 57) | Sandai (n = 59) | Ng'ambo (n = 80) | Salabani (n = 80) | Total (N = 300) |
| Mosquito bite | 7 (29.2) | 11 (19.3) | 14 (23.7) | 30 (37.5) | 25 (31.3) | 87 (29) |
| Contact with arbotus | 1 (4.2) | 2 (3.5) | 3 (5.1) | 1 (1.3) | 2 (2.5) | 9 (3) |
| Contact with infected animals | 8 (33.3) | 19 (33.3) | 25 (42.4) | 28 (35) | 28 (35) | 108 (36) |
| Water contaminated by infected animals at watering points | 3 (12.5) | 9 (15.8) | 17 (28.8) | 23 (28.8) | 21 (26.3) | 73 (24.3) |
| Contact with wildlife | 1 (4.2) | 7 (12.3) | 14 (23.7) | 3 (3.8) | 10 (12.5) | 35 (11.7) |
| Contact with infectious discharge | 1 (4.2) | 2 (3.5) | 13 (22) | 2 (2.5) | 2 (2.5) | 20 (6.7) |
| Other | 3 (12.5) | 6 (10.5) | 8 (13.6) | 11 (13.8) | 20 (25) | 48 (16) |
| **Humans** | | | | | | |
| Contact with infected animal | 0 | 3 (5.3) | 1 (1.7) | 2 (2.5) | 2 (2.5) | 8 (2.7) |
| Contact with infected person | 0 | 0 | 1 (1.7) | 2 (2.5) | 2 (2.5) | 5 (1.7) |
| Drinking unboiled milk from infected animal | 2 (8.3) | 1 (1.8) | 3 (5.1) | 8 (10) | 2 (2.5) | 16 (5.3) |
| Drinking water contaminated by sick animal or person | 0 | 1 (1.8) | 1 (1.7) | 0 | 0 | 2 (0.6) |
| Eating meat and drinking milk from an infected animal, mosquitoes | 0 | 1 (1.8) | 4 (6.8) | 0 | 0 | 5 (1.7) |
| Eating meat and drinking milk from infected animal | 6 (25) | 6 (10.5) | 12 (20.3) | 40 (50) | 35 (43.8) | 99 (33) |
| Eating meat and drinking milk from infected animal, contact with infected animal | 0 | 3 (5.3) | 1 (1.7) | 2 (2.5) | 5 (6.3) | 11 (3.6) |
| Eating meat and drinking milk from infected animal, contact with infected person | 0 | 0 | 1 (1.7) | 2 (2.5) | 2 (2.5) | 5 (1.7) |
| Eating meat and drinking milk from infected animal, contaminated drinking water | 0 | 0 | 1 (1.7) | 0 | 0 | 1 (0.3) |
| Eating meat from infected animal | 3 (12.5) | 13 (22.9) | 9 (15.3) | 12 (15) | 11 (13.8) | 48 (16) |
| Flu and coughing from infected person | 0 | 1 (1.8) | 1 (1.7) | 0 | 1 (1.3) | 3 (1) |
| Mosquito bite | 5 (20.8) | 4 (7) | 2 (3.4) | 1 (1.3) | 3 (3.8) | 15 (5) |
| Contact with wild animals | 0 | 1 (1.8) | 0 | 0 | 0 | 1 (0.3) |

The belief that the mosquito populations had increased or decreased was equally shared among the respondents interviewed. Those that said '*decreased*' gave climate change (less rains) as the main reason for the occurrence of unfavorable conditions for mosquito breeding. On the other hand, those that said '*increased*' gave reasons such as the proliferation of *Prosopis* sp. and *Opuntia* sp., swamps and encroaching lakes and establishment of canals for irrigation.

### Knowledge of Rift Valley fever clinical signs

In humans the commonly mentioned clinical signs of RVF were the classical headache 89/300 (29.7%), fever 75/300 (25%), diarrhoea 71/300 (23.7%), vomiting 70/300 (23.3%) and weakness 61/300 (20.3%) while other symptoms of disease such as bleeding tendencies were rarely mentioned (Fig 3). '*Other*' clinical signs in humans were "*bleeding from the ears*", "*yellowing of eyes*", "*blood in urine*" and loss of weight.

In animals there was a wider spectrum of RVF clinical signs reported (Fig 4). Lack of appetite in animals 58/300 (19.3%) was most frequently mentioned, followed by abortion 57/300 (19%), weight loss 54/300 (18%), bloody diarrhoea 48/300 (16%) and increased salivation 44/300 (14.7%). In animals' '*Other*' clinical signs referred to signs such as "*blood in urine*" and jaundice and were mentioned by a few respondents.

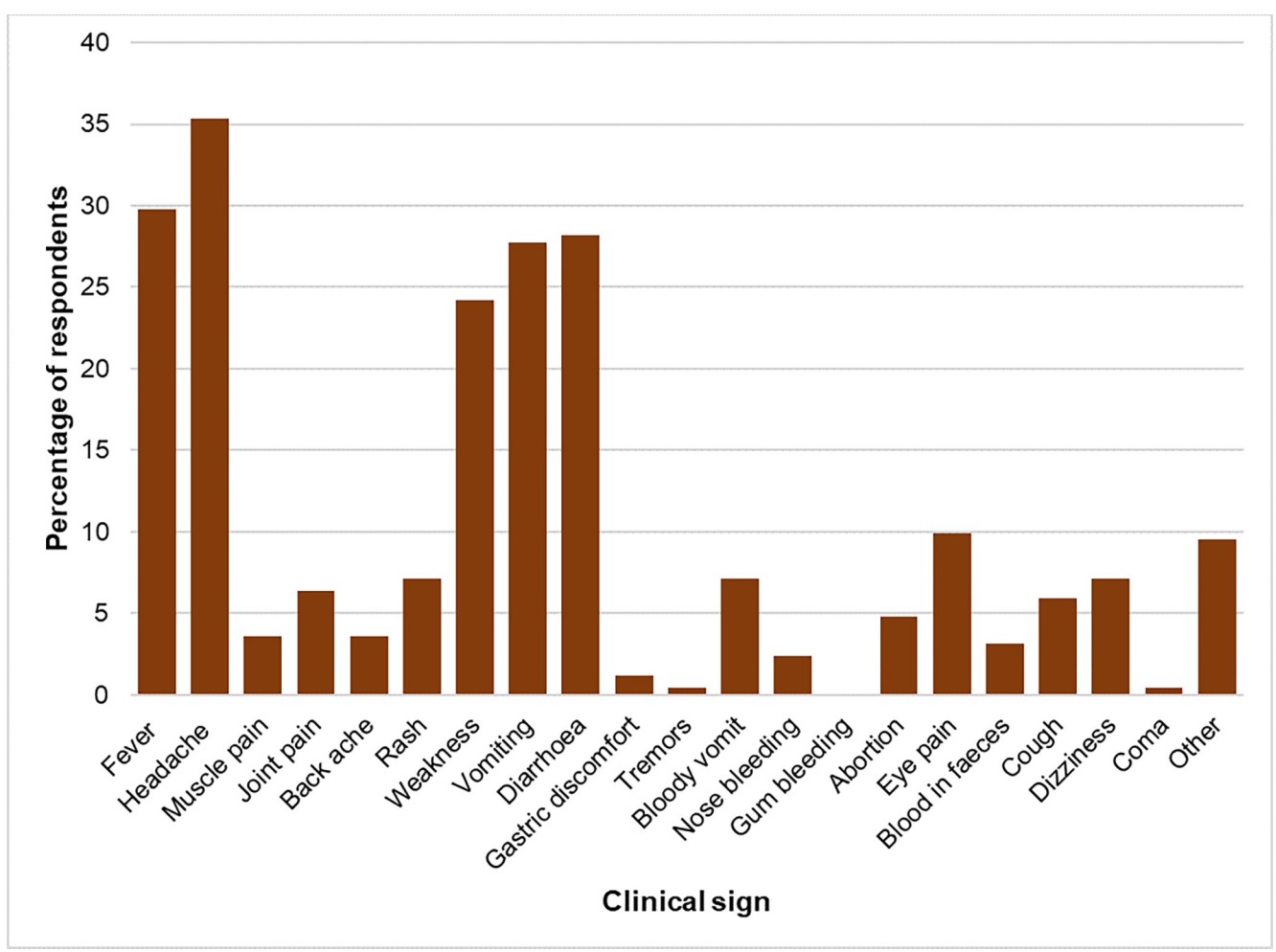

**Fig 3. Community knowledge of Rift Valley fever clinical signs in humans.**

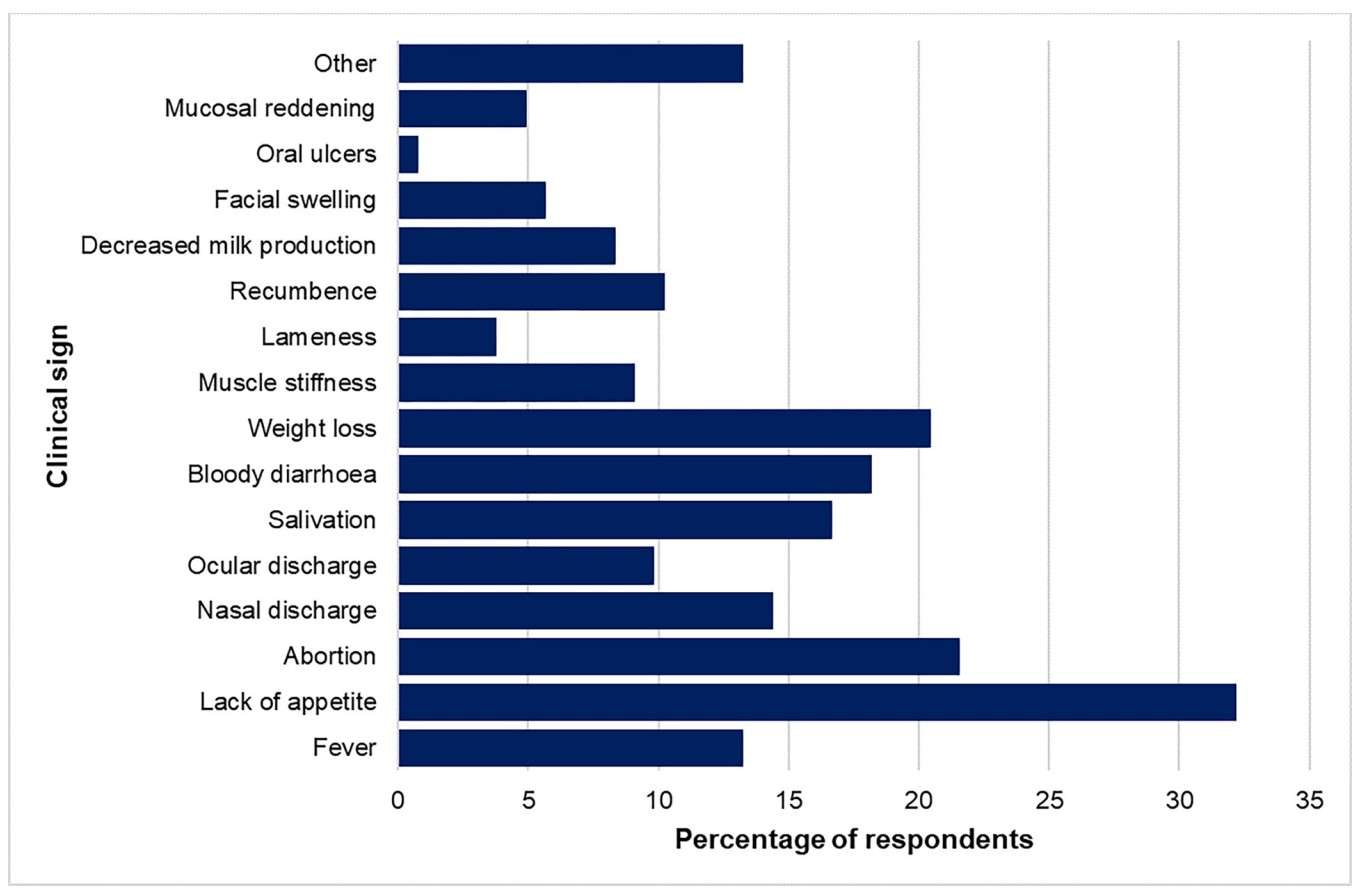

**Fig 4. Community knowledge of Rift Valley fever clinical signs in animals.**

**Knowledge of the risk factors associated with Rift Valley fever infection.** Most of the respondents strongly agreed that consumption of raw/undercooked animal products presented a high risk of contracting the virus (Fig 5). Flooding, bushy vegetation, wildlife presence and contact with animals were also implicated as risk factors in the transmission of the virus. On the other hand, most of the respondents disagreed with the fact that irrigation is a risk factor in the epidemiology of RVF virus.

**Knowledge of the general preventative and current strategies against Rift Valley fever.** Vaccination was the most mentioned 190/300 (63.3%) mode of preventing RVF and outbreaks, followed by the use of bed nets against mosquito bites 135/300 (45%) and hygiene (cooking of meat and boiling of milk) 100/300 (33.3%). Other less frequently mentioned methods are shown in Table 3. Those methods classified as '*Other*' included responses such as, "*burning of herbs to repel mosquitoes*", "*burning cow dung to repel mosquitoes*", "*dipping cattle*", "*taking cattle to where there is no RVF outbreaks*" and "*reducing cattle population*".

Recently, the most commonly mentioned preventative method against RVF was vaccination of animals 203/300 (67.7%) followed by vaccination of people 125/300 (41.7%). These measures were reported to be spearheaded by officials from the National and County governments through veterinary and health professionals with complementary work from non-governmental organizations. Other notable measures carried out by the government were bush clearing 91/300 (30.3%) and insecticide spraying 54/300 (18%). Less commonly mentioned was installation of vector traps, eliminating purported wildlife reservoirs and construction of

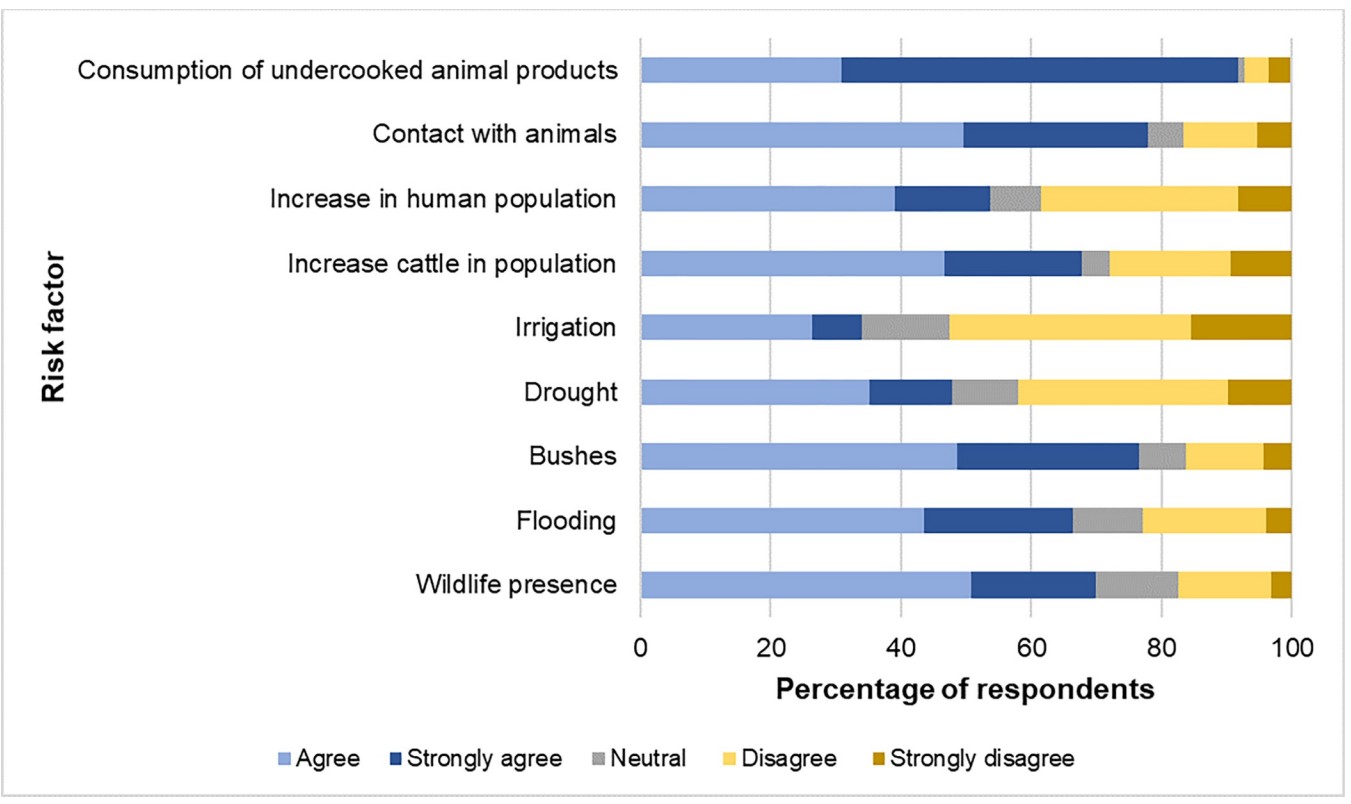

**Fig 5. Community knowledge of the risk factors of Rift Valley fever infection.**

fences to limit livestock-wildlife interaction. Other methods involved draining of stagnant water, isolation of sick animals and awareness campaigns which were mainly attributed to individual and community effort led by community elders. Most of the respondents believed that these measures being carried out are effective for the sole reason that there has not been a recent outbreak of RVF in the area.

The total knowledge score of RVF was significantly associated with age, education, ethnicity, socio-economic index and history of RVF. Households with older heads had more knowledge (p < 0.001) of the disease compared to younger ones (Table 4). Education was significantly associated with knowledge of RVF (p < 0.001). Having formal education up to secondary improved the odds of having better knowledge (OR = 1.28, 95% CI = 1.10–1.49, p = 0.001) compared to those who had gone up to primary level only or not at all. The Maasai had more knowledge on RVF (OR = 1.18, CI = 1.09–1.28, p < 0.001) compared to the Kalenjin. Households with history of RVF in household members (OR = 1.25, 95% CI = 1.10–1.40, p < 0.001) or livestock (OR = 1.38, 95% CI = 1.25–1.52, p < 0.001) had better knowledge of it compared to households without this experience. Respondents from households with a higher socio-economic status had better knowledge (OR = 1.10, CI = 1.01–1.19, p = 0.038) compared to those with less (Table 4).

## Attitudes of respondents to Rift Valley fever threat, practices and occurrence

Most of the respondents had a positive attitude to the threat posed by RVF to human 219/300 (73%) and animal health 255/300 (85%) (Fig 6). The majority 229/300 (76.3%) of the respondents indicated that in case of a suspected infection with the RVF virus they would visit a

**Table 3. Prevention strategies against Rift Valley fever as reported by respondents by location.**

| Method | Frequency (Proportion %) | | | | | |
|---|---|---|---|---|---|---|
| | Kapkuikui (n = 24) | Loboi (n = 57) | Sandai (n = 59) | Ng'ambo (n = 80) | Salabani (n = 80) | Total (N = 300) |
| Reduce contact with sick animals | 5 (20.8) | 5 (8.8) | 12 (20.3) | 9 (11.3) | 12 (15) | 43 (14.3) |
| Use protective equipment when dealing with sick people/animals | 3 (12.5) | 0 | 1 (1.7) | 9 (11.3) | 2 (2.5) | 15 (5) |
| Separate sick animals from healthy ones | 5 (20.8) | 6 (10.5) | 13 (22) | 14 (17.5) | 15 (18.8) | 53 (17.7) |
| Avoid grazing cattle with wildlife | 1 (4.2) | 5 (8.8) | 5 (8.5) | 8 (10) | 6 (7.5) | 25 (8.3) |
| Eliminating wildlife reservoirs | 1 (4.2) | 0 | 3 (5.1) | 1 (1.3) | 1 (1.3) | 6 (2) |
| Vaccination | 15 (62.5) | 30 (52.6) | 39 (66.1) | 51 (63.8) | 55 (68.8) | 190 (63.3) |
| Properly disposing off dead animals/arbotus | 8 (33.3) | 5 (8.8) | 3 (5.1) | 4 (5) | 1 (1.3) | 21 (7) |
| Proper cooking of meat and boiling milk | 8 (33.3) | 10 (17.5) | 13 (22) | 43 (53.8) | 26 (32.5) | 100 (33.3) |
| Bed nets | 11 (45.8) | 18 (31.8) | 30 (50.8) | 38 (47.5) | 39 (48.8) | 136 (45) |
| Install window screens | 0 | 0 | 0 | 1 (1.3) | 1 (1.3) | 2 (0.6) |
| Avoid mosquito areas | 2 (8.3) | 0 | 3 (5.1) | 4 (5) | 4 (5) | 13 (4.3) |
| Avoid grazing livestock in mosquito areas | 0 | 3 (5.3) | 1 (1.7) | 6 (7.5) | 3 (3.8) | 13 (4.3) |
| Use of mosquito repellent | 5 (20.8) | 1 (1.8) | 6 (10.2) | 15 (18.8) | 7 (8.8) | 34 (11.3) |
| Use of mosquito coils | 7 (29.2) | 5 (8.8) | 13 (22) | 13 (16.3) | 6 (7.5) | 44 (14.7) |
| Eliminate mosquito breeding sites | 4 (16.7) | 5 (8.8) | 6 (10.2) | 9 (11.3) | 7 (8.8) | 31 (10.3) |
| Insecticide spraying | 4 (16.7) | 7 (12.3) | 10 (16.9) | 13 (16.3) | 16 (20) | 50 (16.7) |
| Use of mosquito traps | 3 (12.5) | 2 (3.5) | 1 (1.7) | 8 (10) | 6 (7.5) | 20 (6.7) |
| Other methods | 2 (8.3) | 6 (10.5) | 9 (15.3) | 9 (11.3) | 6 (7.5) | 32 (10.7) |
| I don't know | 1 (4.2) | 5 (8.8) | 5 (8.5) | 3 (3.8) | 7 (8.8) | 24 (8) |

health center/hospital to get a proper diagnosis, medication and to be attended to by health professionals as the disease is very serious and needs urgent medical attention. Self-treatment using herbs was also mentioned by a small proportion 12/300 (4%) of the respondents, with the belief that herbs are more potent than conventional medication. Other actions mentioned when suspecting infection included "*do nothing*" ("*It's an animal disease*") (0.6%; 2/300), buy medication from the pharmacy ("*while waiting to get to hospital*"), (0.6%; 2/300) visiting traditional healers (0.3;1/300 and drinking sheep blood mixed with its faeces (0.6%; 2/300). Overall, visiting the hospital was perceived to be very effective in resolving the illness. When suspecting their animals have contracted RVF virus most 163/300 (54.3%) of the respondents indicated that they would call a veterinary officer to assist while another commonly mentioned action was to purchase medicines at an agrovet and self-treat 62/300 (20.7%). Self-treatment with herbal medicines, isolation of sick animals and vaccination were also mentioned. Calling a veterinary officer for assistance was perceived to be the most effective course of action. Burying carcasses was the most mentioned 208/300 (69.3%) mode of disposing animals that die of suspected RVF while other respondents hinted at eating the meat or giving it to their dogs (1.3%; 4/300). A practice that involved mixing the meat with certain herbs before consumption was also mentioned as being done mostly by the elderly in the community.

After fitting a logistic regression model, ethnicity and the socio-economic index were associated with a positive attitude towards RVF (Table 5). The Maasai had a positive attitude (OR = 3.03, 95% CI = 1.42–6.82, p = 0.005) to RVF in comparison to the Kalenjin. Households with a higher socio-economic index had a positive attitude (OR = 2.80, 95% CI = 1.31–6.32, p = 0.01) to RVF compared to those in the lower category.

Most of the respondents highlighted that there seem to be a decrease in the occurrence of the disease in humans 241/300 (80.3%) and animals 246/300 (82%) over the past 10 years. The

**Table 4. Multivariable regression analysis of variables associated with overall knowledge of Rift Valley fever.**

| Variable | Crude OR (95% CI) | p-value | Adjusted OR (95% CI) | p-value |
|---|---|---|---|---|
| *Gender* | | | | |
| Female | 1.028 (0.95–1.12) | 0.513 | | |
| Male | reference | | | |
| *Age* | | | | |
| 60–100 | 1.25 (1.10–1.42) | | 1.31 (1.12–1.53) | < 0.001 |
| 50–59 | 1.18 (1.03–1.35) | | 1.22 (1.05–1.42) | 0.008 |
| 40–49 | 1.42 (1.25–1.62) | | 1.47 (1.29–1.68) | < 0.001 |
| 30–39 | 1.30 (1.14–1.46) | | 1.37 (1.21–1.56) | < 0.001 |
| 19–29 | reference | < 0.001 | | *< 0.001* |
| *Ethnicity* | | | | |
| Maasai | 1.18 (1.09–1.27) | < 0.001 | 1.18 (1.09–1.28) | *< 0.001* |
| Kalenjin | reference | | | |
| *Education level* | | | | |
| Tertiary | 1.0 (0.86–1.16) | | 1.16 (0.96–1.40) | 0.115 |
| Secondary | 1.17 (1.04–1.31) | | 1.28 (1.10–1.49) | 0.001 |
| Primary | 0.92 (0.83–1.01) | | 0.90 (0.82–1.04) | 0.193 |
| None | reference | < 0.001 | | *< 0.001* |
| *History of RVF (human case)* | | | | |
| YES | 1.42 (1.27–1.60) | < 0.001 | 1.25 (1.10–1.40) | *< 0.001* |
| NO | reference | | | |
| *History of RVF (animal case)* | | | | |
| YES | 1.44 (1.32–1.57) | < 0.001 | 1.38 (1.25–1.52) | *< 0.001* |
| NO | reference | | | |
| *Tropical livestock units* | | | | |
| High | 0.94 (0.87–1.02) | 0.131 | | |
| Low | reference | | | |
| *Socio-economic index* | | | | |
| High | 1.08 (1.00–1.16) | 0.062 | 1.10 (1.01–1.19) | *0.038* |
| Low | reference | | | |

OR = odds ratio, CI = confidence interval. *p*-values less than 0.05 in the adjusted model are shown in italics. The independent variables had GVIFs less than 1.2 indicating absence of serious multicollinearity. The Hosmer-Lemeshow goodness of fit test gave a chi-square value of 1.4063, df = 8 and a *p*-value of 0.994 suggesting that the model fits the data well. The area under the curve for the model was 0.74 which is an acceptable predictive power (S1 Fig).

major reasons attributed for this change included vaccination campaigns by the national and county governments, awareness campaigns resulting in improved knowledge of risk practices and prevention methods. Other less mentioned reasons were availability of veterinary/health officers and medication and the advent of climate change that has gradually resulted in less rainfall over the years and decrease in the availability of mosquito habitats.

## Discussion

We carried out a cross-sectional questionnaire survey in the Baringo South sub-County of Kenya, a RVF hotspot which recorded a high number of human and animal cases in the last major outbreak of 2007/2008 [16, 34]. It is therefore unsurprising that most of the respondents had heard about RVF before. However, this study showed gaps in their knowledge as highlighted by the low level of knowledge of RVF transmission, clinical signs, risk factors and prevention. Several socio-economic variables which need to be considered when planning

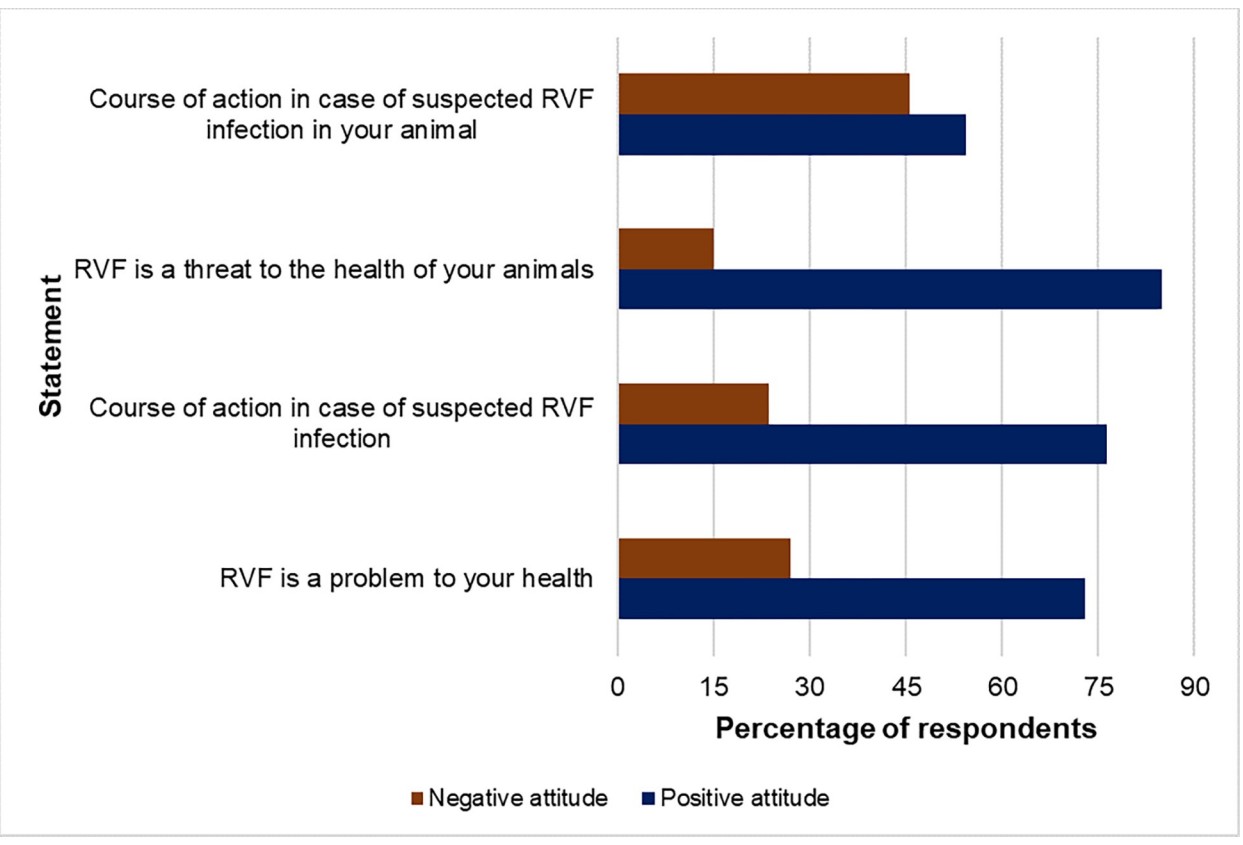

**Fig 6. Community attitude to the threat posed by Rift Valley fever in humans and animals.**

awareness campaigns were associated with knowledge and attitudes to RVF. Unlike most of the questionnaire surveys carried out before, respondents were asked to give their perception on the link there might be, between the changing ecology in their area (invasive plants and conservation) and occurrence of RVF. They were also asked to highlight the current RVF prevention strategies, their implementation and effectiveness.

The knowledge deficit is of concern given that the knowledge of transmission, clinical signs of RVF and its prevention was equally low. Knowledge was, however, high for risk factors of RVF where at least 50% of the respondents got at least half of the total score. The total knowledge score in our study is lower than that recorded in previous studies in Baringo County [15, 20], Ijara district [21], Isiolo County [22] and in Tanzania [10]. This trend can be attributed to a progressive loss of interest associated with a long inter-epidemic period [21] given that the last major RVF outbreak was about 15 years ago [34]. It is therefore important for the County and National governments to keep communities in RVF prone areas informed by carrying out regular awareness campaigns to maintain the knowledge of the disease. Knowledge of RVF is expectedly higher in Isiolo and Ijara because these are pastoralist communities and historical RVF hotspots which are most likely to have benefited from frequent awareness campaigns during previous outbreaks compared to Baringo [21]. In most of the surveys, respondents had a great deal of knowledge on the risk factors of RVF compared to other aspects of the disease most likely because they have an important socio-cultural bearing. For example, people are likely to remember aspects that pertain to proper cooking of meat, discourage drinking unboiled milk/raw blood and disposal of carcasses in a way contrary to their beliefs.

**Table 5. Multivariable regression analysis of variables associated with a positive attitude towards Rift Valley fever.**

| Variable | Crude OR (95% CI) | *p-value* | Adjusted OR (95% CI) | *p-value* |
|---|---|---|---|---|
| Gender | | | | |
| Female | 0.63 (0.32–1.27) | 0.189 | | |
| Male | reference | | | |
| Age | | | | |
| 60–100 | 1.88 (0.75–4.84) | | 2.29 (0.86–6.35) | 0.102 |
| 50–59 | 2.57 (0.92–7.91) | | 3.92 (1.31–13.10) | 0.019 |
| 40–49 | 3.19 (1.16–9.77) | | 3.60 (1.24–11.66) | 0.023 |
| 30–39 | 4.52 (1.57–14.98) | | 6.83 (2.21–24.43) | 0.001 |
| 19–29 | reference | 0.053 | | 0.053 |
| Ethnicity | | | | |
| Maasai | 1.97 (1.00–3.99) | 0.052 | 3.03 (1.42–6.82) | *0.005* |
| Kalenjin | reference | | | |
| Education level | | | | |
| Tertiary | 1.83 (0.43–12.59) | | | |
| Secondary | 1.59 (0.51–5.44) | | | |
| Primary | 0.79 (0.33–1.76) | | | |
| None | reference | 0.405 | | |
| History of RVF (human case) | | | | |
| YES | 6.07 (1.25–109.34) | 0.08 | 6.31 (1.21–116.35) | 0.08 |
| NO | reference | | | |
| Tropical livestock units | | | | |
| High | 0.68 (0.31–1.39) | 0.308 | | |
| Low | reference | | | |
| Socio-economic index | | | | |
| High | 1.86 (0.94–3.88) | 0.083 | 2.80 (1.31–6.32) | *0.01* |
| Low | reference | | | |

OR = odds ratio, CI = confidence interval. *p*-values less than 0.05 in the adjusted model are shown in italics. The independent variables had GVIFs less than 1.1 indicating absence of serious multicollinearity. The Hosmer-Lemeshow goodness of fit test gave a chi-square value of 7.8112, df = 8 and a *p*-value of 0.4521 suggesting that the model fits the data well. The area under the curve for the model was 0.75 which is an acceptable predictive power (S1 Fig).

In our study, non-specific clinical signs of RVF in humans were frequently mentioned compared to specific clinical signs such as bleeding tendencies, abortion and jaundice, corroborating results from previous surveys [20, 21, 35]. While fever is the most known sign of RVF the latter are more important in distinguishing it from other causes of febrile illness such as malaria and other arboviruses in areas where they are co-endemic. Baringo County is prone to seasonal transmission of malaria during the rainy season [36]. It is therefore important to put more emphasis on clinical signs that may be less common but can be better used to identify suspected RVF cases early into an outbreak [37]. In several cases during the questionnaire interviews it seemed what some respondents were referring to as RVF could have been yellow fever whose epidemiology and clinical syndrome is quite similar to RVF.

While the knowledge of RVF transmission routes was also low the major routes were frequently mentioned. Some respondents highlighted that the virus could be transmitted between humans through contact and *via* coughing; however, person to person transmission of RVF has not been demonstrated [38]. Dispelling these incorrect notions about the disease is important to avoid unwarranted stigmatization of infected people. None of the respondents were able to name the mosquito species responsible for RVF transmission with only a few

mentioning mosquitoes as being important in the transmission of RVF which indicates low knowledge of the vector and the epidemiology of the disease. Most of the education on RVF has not emphasized on this important aspect of the epidemiology of the disease as much as the *Anopheles* mosquito in malaria transmission. Low level of knowledge on the role of mosquitoes in RVF epidemiology has been evident in previous studies in Kenya [20, 21], Uganda [37], Malawi [39] and Tanzania [10].

Knowledge of risk factors for RVF generally leads to avoidance behavior and preparedness during outbreaks. It is important to note that most of the respondents in our study agreed/ strongly agreed that contact with animals, their secretions and products are important risk factors for RVF and generally for any zoonotic pathogen. These findings are similar to a study in Tanzania where about 73% of the respondents were aware of these risk factors [10]. While knowledge of risk factors entails correct implementation of preventative measures it is hampered by cultural behaviors such as drinking raw milk/blood, especially among pastoralist communities in Kenya [15, 21, 40] and Tanzania [35]. While the shedding of the virus into milk is thought to be low, blood has been reported to be viraemic and highly infectious [6]. Presence of bushy vegetation and flooding are also important risk vectors for RVF outbreaks. Flooding results in the hatching of *Aedes* spp. eggs which give rise to infective mosquitoes that transmit the virus to nearby hosts [34]. It's important to note that most of the respondents in our study believe that the invasive plants *Prosopis* spp. and *Opuntia* spp. which are widespread in Baringo County harbor mosquitoes which bite them. Bushy vegetation generally provides shaded resting places for adult mosquitoes while the litter creates conducive mosquito oviposition microhabitats. Massive presence of these plants is thought to influence plant feeding behavior of mosquitoes and subsequently their vector competence [23, 41].

The majority of the respondents disagreed that irrigation could be a risk factor. In Baringo most of the agriculture is practiced close to rivers and lakes to facilitate flood irrigation. This generally results in artificial prolonged water stagnation which can support the breeding of mosquitoes. In a previous study in the Kenyan Tana River County, more mosquitoes, especially primary RVF vector species, were collected in flood irrigated farms and villages compared to rainfed agricultural land [42]. While the belief by our respondents that wildlife in the community conservancies presents high risk for RVF to their livestock is common, its role in the maintenance of enzootic circulation and amplification during inter-epidemic periods is poorly understood [43].

Vaccination as a method of preventing RVF outbreaks was frequently mentioned in our study with communities in previous studies also showing a positive attitude towards it [21]. While the role of livestock vaccination in Kenya is quite clear in preventing RVF outbreaks, the mention of vaccination against RVF in humans by some respondents could have been erroneous as the human vaccine is not available on the market. It is possible that some of the vaccination campaigns were not for RVF but for other pathogens in the communities such as yellow fever or Coronavirus disease. The major players in all RVF vaccination campaigns have been the national and county governments as acknowledged by most of the respondents. This is expected as RVF is a notifiable disease in Kenya and its control thus a responsibility of the government. Both veterinary and health professionals were frequently mentioned together as the key personnel driving the government initiative to prevent RVF outbreaks in the area indicating a '*One health*' approach to the problem. This is an integrated and interdisciplinary approach to achieve the health of humans, domestic and wild animals, plants, and the wider environment [44].

Vector control-based preventative methods were not well known except the use of mosquito bed nets. This, however, implies low level of knowledge on the bionomics of the vector as *Aedes* spp. mosquitoes rarely bite indoors and at night [45]. Majority of the respondents in

this study highlighted that the control measures implemented by the government, especially vaccination campaigns, have been highly effective as demonstrated by the absence of any RVF outbreaks lately. Therefore, the belief by at least 80% of the respondents is that the occurrence of the disease has decreased or virtually '*disappeared*'. This belief could probably explain the low level of knowledge in this study as highlighted earlier. It is also not entirely true because outbreaks occur irregularly and inter-epidemic periods maybe as long as 15 years [12]. Moreover, inter-epidemic circulation can still occur resulting in illness in animals and people; therefore, they should remain wary of the threat posed by RVF [46, 47].

At least 70% of the respondents had a positive attitude towards the threat posed by RVF in both animals and humans and indicated that in suspected cases they would seek advice from veterinary professionals and health centers, respectively. Similar findings were previously reported in Kenya [20] and Tanzania [10]. This positive attitude is most likely to stem from the morbidity and mortality due to RVF witnessed in these communities during the last outbreak in 2007/2008. A few respondents indicated that they use traditional herbs to treat themselves and their livestock during outbreaks, however, the herbs were not specified. This is similar to other studies where some livestock keepers indicated that they use herbal medicines to treat their animals after they aborted [15]. Some pastoralists believed that home remedies, such as mixtures of raw blood, milk and honey, could cure RVF as they did not have faith in conventional hospitals [48]. While the most frequently method of disposing of animals that have died of suspected RVF was burying the carcass without opening it, a few respondents indicated they give the meat to their dogs or even consume it, thereby exposing themselves to risk of contracting the virus. In another study in Baringo herbs from *Acacia* spp. were used to make meat from RVF affected carcasses '*safe*' for consumption. This is based on some cultural beliefs that an animal should not be buried but consumed [15]. Despite these behaviors indicating a rich base of indigenous knowledge in communities, their efficacy should be analysed first before widespread acceptance. Several plant extracts have actually shown *in vitro* anti-RVF virus activity [49]. However, these practices can be unsafe and expose communities to risk of infection during outbreaks. Self-treatment with medicines bought from pharmacies and agrovets was also mentioned in this study just like in an earlier study from Baringo County [15]. This should also be discouraged as there is no cure for RVF and treatment involves only supportive care.

Households with older heads had more knowledge on RVF, probably because of exposure to several RVF outbreaks and awareness campaigns. Also, they are more likely to be involved and depend on their livestock compared to the younger generation who may have other sources of income. Education level higher than primary school was associated with better knowledge on RVF mainly because education gives one the ability to easily comprehend information about a disease [50, 51]. The relationship between education level and knowledge of RVF has been reported before in Baringo [20] and Isiolo Counties [22]. A higher socio-economic index, reflecting the wealth status of a household, was associated with better knowledge and a positive attitude to RVF because wealthy households are most likely to have more access to information on diseases and their prevention and can afford to manage the disease [20, 31]. Households that had a history of RVF exhibited better knowledge of the disease compared to those who did not because this past experience which may have resulted in passing of a household member or loss of livestock remains etched in their memory [20, 21]. The Maasai were more knowledgeable and had a more positive attitude to RVF compared to the Kalenjin, possibly because they are more of livestock keepers with greater experiences in managing livestock diseases while the Kalenjins are more of agro-pastoralists. In previous studies, age and education level were not associated with knowledge and attitude [10, 21] signifying that these associations differ according to regions and should not be generalised during design of awareness campaigns.

Our respondents rated radio and television as the most effective method for disseminating information on RVF, similar to previous studies [21]. Radio is especially important as it is easily accessible even to illiterate members of the community. However, community gatherings are also still a crucial portal for dissemination of RVF information [21] while the use of social media could also be advocated for because most of the younger people are on these platforms. The findings of this study should be construed in light of the following limitations: The last major RVF outbreak that affected Baringo County occurred about 13 years ago and hence the study could have been affected by recall bias in the participants. While it is important for the participants to have good knowledge on RVF it is equally important to translate the knowledge into good practices. Some bias could be introduced when this '*good practice*' is self-reported as in this study rather than observed. It is a challenge to accurately determine if there is history of RVF occurrence in the household (human or livestock cases) as some cases may have been suspected but not diagnosed by a clinician or veterinarian. Nonetheless this study provides a useful perspective on the current level of knowledge on RVF and its determinants in Baringo County paving way for remedial action.

## Conclusion and recommendations

There was low level of knowledge on RVF in Baringo South sub-County, a known hotspot of the last major RVF outbreak in Kenya in 2007/2008. Out of the four scales considered (transmission, clinical signs, risk factors and prevention) respondents had better knowledge on risk factors only. However, the majority had a positive attitude towards RVF. Consequently, there is need to regularly carry out education/awareness campaigns that focus on these aspects of the disease. In addition to community gatherings, radio and television are major sources for information dissemination. Low knowledge of the vectors of RVF calls for greater emphasis on this aspect and the link between mosquitoes, their ecology and outbreaks of RVF should be clearly explained to communities.

The high knowledge of risk factors of RVF is encouraging. However, some practices and beliefs that discourage seeking for professional assistance in suspected cases while promoting consumption of carcasses should be firmly discouraged especially during outbreaks. The level of knowledge and attitude to RVF in the interviewed households was influenced by age, education and history of RVF. Community elders with more experience with RVF are therefore important in imparting knowledge to the younger generation. This can be done at community gatherings where other members of the community who have history of RVF share their experiences. The purported role of wildlife in the maintenance of RVF during IEPs, especially in locations close to conservation areas, should be clearly explained to the communities as they often seem to have a negative attitude towards these entities. This is also the case with invasive plants such as *Prosopis* spp. and *Opuntia* spp. whose role in mosquito survival and competence is still under investigation.

## Supporting information

**S1 Questionnaire. Arbovirus questionnaire.**
(PDF)

**S1 Text. PLOS inclusivity in global research checklist.**
(DOCX)

**S2 Text. Socio-economic scores and contributions of household characteristics and assets to multiple correspondence analysis (MCA).**
(DOCX)

**S1 Fig. Receiver operating curves (ROC) for the knowledge and attitude models.** (TIF)

## Acknowledgments

We are grateful for the input and assistance of Dr. Léa Lacan and Wisse van Engelen (both University of Cologne and Paul Nyangau (*icipe*) in drafting of the questionnaire. We also thank Dr. Dishon Muloi (ILRI) for assistance with the statistical analysis and Dr. Bester T. Mudereri (*icipe*) for assistance in producing the study site map. In addition, we acknowledge the assistance rendered by the Kenya Forestry Research Institute (KEFRI) in Marigat for logistical support during the survey. We also acknowledge the support from the CGIAR One Health initiative "Protecting Human Health Through a One Health Approach", which was supported by contributors to the CGIAR Trust Fund (https://www.cgiar.org/funders/)). We also thank the enumerators, the communities in Baringo South and their leaders for allowing us to carry out this research in their area.

## Author Contributions

**Conceptualization:** Sandra Junglen, Christian Borgemeister.

**Data curation:** Tatenda Chiuya.

**Formal analysis:** Tatenda Chiuya.

**Funding acquisition:** Sandra Junglen, Christian Borgemeister.

**Investigation:** Tatenda Chiuya, Eric M. Fevre.

**Methodology:** Tatenda Chiuya, Eric M. Fevre, Sandra Junglen, Christian Borgemeister.

**Project administration:** Tatenda Chiuya, Eric M. Fevre, Sandra Junglen, Christian Borgemeister.

**Resources:** Eric M. Fevre, Sandra Junglen, Christian Borgemeister.

**Software:** Tatenda Chiuya.

**Supervision:** Eric M. Fevre, Sandra Junglen, Christian Borgemeister.

**Validation:** Tatenda Chiuya.

**Visualization:** Tatenda Chiuya.

**Writing – original draft:** Tatenda Chiuya.

**Writing – review & editing:** Tatenda Chiuya, Eric M. Fevre, Sandra Junglen, Christian Borgemeister.

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
