## [Decision Letter · Decision Letter 0]

15 Jun 2023

PGPH-D-23-00614

Understanding knowledge, attitude and perception of Rift Valley fever in Baringo South, Kenya: A cross-sectional study

Dear Dr. Chiuya,

Thank you for submitting your manuscript to PLOS Global Public Health. After careful consideration, we feel that it has merit but does not fully meet PLOS Global Public Health’s publication criteria as it currently stands. Therefore, we invite you to submit a revised version of the manuscript that addresses the points raised during the review process.

We look forward to receiving your revised manuscript.

Kind regards,

Sukanta Chowdhury, Ph.D

Academic Editor

Journal Requirements:

1. Please include a complete copy of PLOS’ questionnaire on inclusivity in global research in your revised manuscript. Our policy for research in this area aims to improve transparency in the reporting of research performed outside of researchers’ own country or community. The policy applies to researchers who have travelled to a different country to conduct research, research with Indigenous populations or their lands, and research on cultural artefacts. The questionnaire can also be requested at the journal’s discretion for any other submissions, even if these conditions are not met.  Please find more information on the policy and a link to download a blank copy of the questionnaire here: https://journals.plos.org/globalpublichealth/s/best-practices-in-research-reporting. Please upload a completed version of your questionnaire as Supporting Information when you resubmit your manuscript.”

b. If any authors received a salary from any of your funders, please state which authors and which funders.

3. In the online submission form, you indicated that "The dataset generated and analysed in this study can be made available from the corresponding authors on reasonable request". All PLOS journals now require all data underlying the findings described in their manuscript to be freely available to other researchers, either 1. In a public repository, 2. Within the manuscript itself, or 3. Uploaded as supplementary information.

Additional Editor Comments (if provided):

Reviewers' comments:

Reviewer's Responses to Questions

**Comments to the Author**

1. Does this manuscript meet PLOS Global Public Health’s publication criteria? Is the manuscript technically sound, and do the data support the conclusions? The manuscript must describe methodologically and ethically rigorous research with conclusions that are appropriately drawn based on the data presented.

Reviewer #1: Yes

Reviewer #2: Yes

Reviewer #3: Yes

2. Has the statistical analysis been performed appropriately and rigorously?

Reviewer #1: No

Reviewer #2: I don't know

Reviewer #3: Yes

3. Have the authors made all data underlying the findings in their manuscript fully available (please refer to the Data Availability Statement at the start of the manuscript PDF file)?

Reviewer #1: Yes

Reviewer #2: Yes

Reviewer #3: No

4. Is the manuscript presented in an intelligible fashion and written in standard English?

Reviewer #1: Yes

Reviewer #2: No

Reviewer #3: Yes

5. Review Comments to the Author

Reviewer #1: The authors investigated risk factors of RVF using a structured questionnaire. Major and minor

concerns are listed below.

1. In line 121, it was stated that the “participants were selected randomly”. Please specify which

random sampling was applied during the study.

2. In line 121-122, “The total number of respondents was proportionally constituted based on the

population and number of households in each location.” It is unclear here. Please briefly describe

how it proportionally constituted to the population as well as number of households in each

location.

3. line 124-125, “At each household the head or another person from that household above the

age of 18 were interviewed and data were recorded”. How many people were interviewed from

each household? If more than one person, the why only head of the household were interviewed

not all the participants? It is a knowledge, attitude and perception study, therefore, if only the

head of the household was interviewed then there would be bias in this study.

4. No information regarding the correlation analysis between the continuous independent

variables was mentioned in the data analysis section. Did the author try to find out the correlation

matrix among continuous independent variables? If yes, then what was cut off value to identify

high correlated variables, and what was the basis of selection of variables from highly correlated

variables?

5. The author did not analyze the univariable association between independent and dependent

variables. Is there any specific reason for not doing univariable analysis?

6. After fitting the final model in multivariable analysis, did the author try to find out if there was

any confounder or not? How was the confounder checked in the final model?

7. There was no information regarding the interaction of the variables from the final model. Did

the author check any interactions of significant variables from the final model?

8. How the goodness of the fit of final model was checked? Please specify which test was used to

find out the overall goodness of fit of the final model.

9. How the predictive ability of the model was identified? Generally, the Receiver Operating

Characteristics (ROC) curve helps to identify the power of the final model. Please add this to the

manuscript.

Reviewer #2: Dear Authors,

I hope this comment finds you well. I recently had the opportunity to review your research article. I have a few suggestions that I believe can enhance the readability and clarity of your manuscript.

Firstly, I would recommend structuring the abstract according to the traditional format of background, methods, results, and conclusion. This standard organization can greatly improve the overall readability of the abstract, allowing readers to quickly grasp the key aspects of your research.

Secondly, I suggest reorganizing the presentation of your results to enhance clarity and flow. By ensuring a logical progression and grouping related findings together, readers will find it easier to follow your research and understand the implications of your results.

Additionally, I encourage you to consider rephrasing sentences throughout the manuscript to achieve greater clarity and conciseness. For example, in sentence 41, you can rephrase it as "Rift Valley fever (RVF) is a zoonotic disease caused by the Rift Valley fever virus (RVFV)." Shortening sentences will make the content more accessible and facilitate a better understanding of the subject matter.

Overall, your research demonstrates significant potential, and addressing these minor improvements can make your manuscript more accessible and impactful to a wider audience. I appreciate the effort you have put into this study and believe that these suggestions will enhance the overall quality of your work.

Thank you for considering my comments, and I look forward to seeing your revised manuscript.

Best regards

Tirth Dave

Reviewer #3: Abstract: “86% had a positive attitude”

- I know this is just an abstract so hard to elaborate, but would help to add little more info so the reader knows what “positive attitude” means. They like RVF? They are confident they understand it?

Line 46: In the first sentence of the abstract you say RVF causes high morbidity and mortality in humans but in intro line 46 you say it progresses to sever symptoms “in rare cases.” This feels contradictory to describe it in both ways.

Line 55: “signifying increased receptivity.” The word receptivity is confusing wording, as it suggests acceptance of the virus. I would recommend replacing with something like “signifying increased potential for transmission in most parts of the country.”

Line 62: Most people don’t know what solanchak soils are. Maybe include a brief description of why that type of soil is conducive to RVF outbreaks?

Line 119: “In each of the locations, eight villages were randomly selected with four villages each per sub-location.” I do not understand this sentence. Were 8 villages randomly selected, and then 4 of those 8 were decided on? What are the sub-locations? It would be helpful if this were more clear.

Line 141: I personally recommend against defining acronyms/abbreviations that are not extremely well established and understood by the general field. While writing TLU helps with word count, most readers will forget what that means and you will lose impact when reporting the results. For example, while I was reading I got lost trying to go back up to find the definition, and then go back to where I was reading. If make the change then you’ll need to go through and change TLU to tropical livestock units throughout.

Line 142: As above, I recommend just writing out socio-economic index instead of SEI; it makes life a lot easier for the reader.

Line 145: I am unfamiliar with this specific SEI index and the method (MCA) used to calculate the index. I think it would be really helpful if the authors added a brief explanation of how this works. Are there weights assigned to each of the variables, and is the weight determined by the authors or is weight determined by MCA through some sort of regression?

Line 159: This is the first mention of attitude and it is unclear what attitude means and why attitude is being explored. It is very clear to me why knowledge is important, but not clear my attitude is important. It would be nice if there were a sentence or two explaining the importance of attitude in preventing RVF outbreaks.

Line 179-181: “Most of the household heads were aged between 30-39 years 71/300

180 (23.7%), had an education up to primary level 149/300 (49.7%), were married 231/300 (77%)

181 and protestants 181/300 (60.3%).” The grammar in this sentence makes it hard to understand what numbers are associated with what variables. Can you please revise?

Line 189-191: are relatives considered extended family or does that include friends/neighbors too? If it includes neighbors/friends, maybe just say “12% reported knowing someone who had RVF.” Also there is a difference between having someone in the household with previous exposure and just knowing someone. Maybe report them separately; if someone in their house had been sick they probably are going to be much more aware compared to just knowing someone from a neighboring village.

Line 194: When you report Tropical Livestock Units, I recommend saying whether they had 4 or more, or less than 4, given the overall average was 3.6.

Line 227: Why are the numbers not reported for each location? If you can’t share the data in the manuscript for fear of demeaning these communities, it would be helpful to know if knowledge regarding each topic was consistent across the locations. For example, you could add the range of values across locations to show that some places differ in levels of understanding. Would be very helpful in real life for stakeholders to have this info of course.

Line 235-237: The invasive species question is very interesting. Can you add data on how many implicated this type of vegetation? (not required)

Line 292: Before going into results from the adjusted multivariate regression model, need to explain which covariates were selected for inclusion. At least, covariate selection was what I was expecting based on the statement in lines 169-171.

Line 305: Again, it is confusing what a positive attitude means. This should be explained in the methods.

Line 313-316: add (N=xx) for each of these. Do a considerable proportion of people believe they should mix sheep blood with faeces? This would be concerning. OR is this just a single person?

Line 323-324: what proportion hinted at eating the animals?

Line 331-336: this will be much more clear and interpretable when you explain what positive attitude means.

Line 442: can you please add a citation for timing and location that bites occur

Line 349: I don’t see any discussion about the multivariate model results. This leads to an important question: what do we learn from running a multivariate model that we don’t learn from univariate analysis? In the methods it would be nice to have an explanation of why the decision was made to do an adjusted model, and then add discussion as necessary based on your justification. (Important)

6. PLOS authors have the option to publish the peer review history of their article (what does this mean?). If published, this will include your full peer review and any attached files.

**Do you want your identity to be public for this peer review?** For information about this choice, including consent withdrawal, please see our Privacy Policy.

Reviewer #1: **Yes: **Biswajit Bhowmick

Reviewer #2: **Yes: **Tirth Dave

Reviewer #3: No

---

## [Decision Letter · Decision Letter 1]

4 Aug 2023

PGPH-D-23-00614R1

Understanding knowledge, attitude and perception of Rift Valley fever in Baringo South, Kenya: A cross-sectional study

Dear Dr. Tatenda Chiuya,

Thank you for submitting your manuscript to PLOS Global Public Health. Few minor additional comments are given to address. Therefore, we invite you to submit a revised version of the manuscript that addresses the points raised during the review process.

We look forward to receiving your revised manuscript.

Kind regards,

Sukanta Chowdhury, Ph.D

Academic Editor

Journal Requirements:

Additional Editor Comments:

Abstract: You have mentioned “only 9.6% attained at least half of the overall knowledge score on RVF”. The term “overall knowledge score” is poorly measurable and understandable. I think that it “total score”. Please clarify this.Abstract: You suggested to perform regular awareness campaigns during inter-outbreak period. It is better to perform awareness campaigns regular basis all the time. During outbreak period it is also important.In the multivariate logistic regression, the association between overall knowledge of RVF and factors was identified. It is important to define the “overall knowledge” (e.g. participants who scored >5 out of 10).Table 4: why the variable “Tropical livestock units” was not included in multivariate logistic regressionTable 5: why the variable “Gender” was not included in multivariate logistic regressionLine number 419: references [6,34] stands for what information?I believe that every study has some limitations. It is important to mention limitation at the end of discussion section if the study had few limitations.

Reviewers' comments:

Reviewer's Responses to Questions

**Comments to the Author**

1. If the authors have adequately addressed your comments raised in a previous round of review and you feel that this manuscript is now acceptable for publication, you may indicate that here to bypass the “Comments to the Author” section, enter your conflict of interest statement in the “Confidential to Editor” section, and submit your "Accept" recommendation.

Reviewer #1: All comments have been addressed

Reviewer #3: All comments have been addressed

2. Does this manuscript meet PLOS Global Public Health’s publication criteria? Is the manuscript technically sound, and do the data support the conclusions? The manuscript must describe methodologically and ethically rigorous research with conclusions that are appropriately drawn based on the data presented.

Reviewer #1: Yes

Reviewer #3: Yes

3. Has the statistical analysis been performed appropriately and rigorously?

Reviewer #1: Yes

Reviewer #3: Yes

4. Have the authors made all data underlying the findings in their manuscript fully available (please refer to the Data Availability Statement at the start of the manuscript PDF file)?

Reviewer #1: Yes

Reviewer #3: Yes

5. Is the manuscript presented in an intelligible fashion and written in standard English?

Reviewer #1: Yes

Reviewer #3: Yes

6. Review Comments to the Author

Reviewer #1: accept

Reviewer #3: This a a much improved version of the manuscript and the work presented will be beneficial for preventing future RVF incidence.

7. PLOS authors have the option to publish the peer review history of their article (what does this mean?). If published, this will include your full peer review and any attached files.

**Do you want your identity to be public for this peer review?** For information about this choice, including consent withdrawal, please see our Privacy Policy.

Reviewer #1: **Yes: **Biswajit Bhowmick

Reviewer #3: No

---

## [Editor Report · Decision Letter 2]

16 Aug 2023

Understanding knowledge, attitude and perception of Rift Valley fever in Baringo South, Kenya: A cross-sectional study

PGPH-D-23-00614R2

Dear Dr. Tatenda Chiuya,

We are pleased to inform you that your manuscript 'Understanding knowledge, attitude and perception of Rift Valley fever in Baringo South, Kenya: A cross-sectional study' has been provisionally accepted for publication in PLOS Global Public Health.

Best regards,

Sukanta Chowdhury, Ph.D

Academic Editor